# Saponin Esculeoside A and Aglycon Esculeogenin A from Ripe Tomatoes Inhibit Dendritic Cell Function by Attenuation of Toll-like Receptor 4 Signaling

**DOI:** 10.3390/nu16111699

**Published:** 2024-05-30

**Authors:** Jian-Rong Zhou, Shigenori Kinno, Kenta Kaihara, Madoka Sawai, Takumi Ishida, Shinji Takechi, Jun Fang, Toshihiro Nohara, Kazumi Yokomizo

**Affiliations:** 1Faculty of Pharmaceutical Sciences, Sojo University, Kumamoto 860-0082, Japanstakechi@ph.sojo-u.ac.jp (S.T.); fangjun@ph.sojo-u.ac.jp (J.F.); none@ph.sojo-u.ac.jp (T.N.); yoko0514@ph.sojo-u.ac.jp (K.Y.); 2School of Pharmacy at Fukuoka, International University of Health and Welfare, Fukuoka 831-8501, Japan; msawai@iuhw.ac.jp (M.S.); ishida@iuhw.ac.jp (T.I.)

**Keywords:** dendritic cell, fresh tomato fruit, Esculeoside A, Esculeogenin A, TLR4, p-NFκB

## Abstract

Dendritic cells (DCs) can initiate immune response through the presenting antigens to naïve T lymphocytes. Esculeoside A (EsA), a spirosolane glycoside, is reported as a major component in the ripe fruit of tomato. Little is known about the effect of tomato saponin on mice bone marrow-derived DCs. This study revealed that EsA and its aglycon, esculeogenin A (Esg-A), attenuated the phenotypic and functional maturation of murine DCs stimulated by lipopolysaccharide (LPS). We found that EsA/Esg-A down-regulated the expression of major histocompatibility complex type II molecules and costimulatory molecule CD86 after LPS stimulation. It was also determined that EsA-/Esg-A-treated DCs were poor stimulators of allogeneic T-cell proliferation and exhibited impaired interleukin-12 and TNF-α production. Additionally, EsA/Esg-A was able to inhibit TLR4-related and p-NFκB signaling pathways. This study shows new insights into the immunopharmacology of EsA/Esg-A, and represents a novel approach to controlling DCs for therapeutic application.

## 1. Introduction

Dendritic cells (DCs), important professional antigen-presenting cells (APCs), are a subpopulation of myeloid-derived leukocytes [1,2,3]. Immature DCs lie in non-lymphoid tissues, can capture antigen, and process it. When there are inflammatory and infectious signals, they migrate into the T-cell zone of lymphoid tissues and mature, acquiring the ability to present antigens to naïve T lymphocyte. Mature DCs show high expression of surface costimulatory molecules CD80 and CD86 and major histocompatibility (MHC)-peptide complexes as maturation markers, and can produce inflammatory cytokines, for example IL-12, IL-1β and tumor necrosis factor α (TNF-α) [4,5,6]. In the intracellular pathway, when DCs are stimulated by lipopolysaccharide (LPS), a membrane glycolipid of Gram (-) bacteria, Toll-like receptor 4 (TLR4) is predominantly bound [7]. This triggers the assembly of a submembranous signaling complex containing an adaptor protein (myeloid differentiation primary response gene 88; MyD88), a protein kinase (IL-1 receptor associated kinase), and TNF receptor-associated factor 6 (TRAF6) [8,9,10,11], which induces IκB kinase (IKK) activation. Following κB (IκB) inhibitor degradation by IKK, the transcription factor nuclear factor κB (NFκB) translocates into the nucleus, then it regulates the expression of proinflammatory cytokine genes [12]. Thus, with the functional transition of DCs from antigen capturing to presenting, DCs not only induce immunogenicity but also exhibit tolerogenicity [13]. The balance between these two states seems to be dependent on DC activation [14]. Allergic diseases, arthritis, and autoimmune disorders could occur [15,16,17] if this balance is disrupted.

Esculeoside A (EsA), a spirosolane glycoside, is isolated from tomato fruit (*Lycopersicon esculentum*) and is also reported as a main component in ripe tomatoes [18], with a four-fold higher content than that of lycopene [19]. Our previous studies reported that EsA ameliorated experimental atopic dermatitis in mice and showed that EsA and its sapogenol esculeogenin A (Esg-A) suppress T-cell activation by modulation of T helper cell (Th2), Th1, and regulatory T-lymphocyte (Treg) differentiation in vitro [20,21]. However, atopic dermatitis lesions were characterized not only by expanded type 2 T cells but also inflammatory DCs [22].

In this study, we tested whether EsA and its aglycon Esg-A affect the LPS-stimulated maturation of DCs derived from murine bone marrow. We show, for the first time, that EsA and Esg-A inhibit the LPS-stimulated phenotypic and functional maturation of DCs, suppress the activation of TLR4, MyD88, and TRAF6 signaling, and also decrease phosphorylated NFκB (p-NFκB) expression in DCs.

## 2. Materials and Methods

### 2.1. EsA and Esg-A Extraction

EsA and its aglycon Esg-A were extracted and isolated as previously described [21,23]. Briefly, cherry tomato fruits (*Lycopersicon esculentum var. cerasiforme*; 2.1 kg (Youyou, Kumamoto, Japan)) were crushed, supplemented with H_2_O and centrifuged at 2600× *g* for 10 min. The supernatant liquid was passed through a Diaion^®^ HP-20 column (Mitsubishi Chemical, Tokyo, Japan) and separated with 40% aq. MeOH, 60% aq. MEOH, and MeOH, successively. EsA was provided in the 60% eluate (1698 mg). EsA (1330 mg) was hydrolyzed with 2 N HCl, and then the reaction solution was extracted with AcOEt. The upper layer was evaporated to afford a residue, which was purified by a silica gel column with CHCl_3_–MeOH–H_2_O = 9:1:0.1 to obtain Esg-A (3.7 mg) [18,22]. The EsA chemical structure is shown in Figure 1A. EsA or Esg-A was prepared and diluted with dimethyl sulfoxide (DMSO) from a stock solution of 300 or 30 μM.

### 2.2. Animals

The present study was permitted by the Sojo University Ethics Committee (2021-P-025, 2022-P-006). All experiments were performed according to the Guidelines about the Care and Use of Laboratory Animals from the Japanese Pharmacological Society. The 6–9-week-old female BALB/c and C57BL/6 mice were purchased from Japan SLC (Hamamatsu, Japan). The animals were housed for at least 1 week before use under 24.5–25.0 °C, humidity (60 ± 10%), and a light/dark cycle. Mice standard chow (SLC, Hamamatsu, Japan) was provided ad libitum.

### 2.3. DC Generation from Murine Bone Marrow

The DCs were generated as per the previous report [24]. Briefly, the mice were sacrificed by using isoflurane, then bone marrow of the femora and tibiae of the BALB/c mice were flushed and passed through a 70-μm filter, then depletion of the red blood cells (RBCs) was carried out using an RBC lysis buffer. The cells were grown from precursors in RPMI 1640 medium (Wako, Osaka, Japan) supplemented with heat-inactivated fetal bovine serum (FBS, 10% (*v*/*v*)), 100 U/mL of penicillin, streptomycin (Invitrogen, Waltham, MA, USA), and 2-mercaptoethanol. Recombinant mouse granulocyte–macrophage colony-stimulating factor (10 ng/mL, GM-CSF) (R&D Systems, Minneapolis, MN, USA) was added, subsequently referred to as the complete medium. After a 1-week culture, non-adherent and loosely adherent cells were collected as DCs, adjusted to 2 × 10^5^ cells/mL and seeded onto 24-well plates. After exposure to EsA or Esg-A for 1 h, the DCs were stimulated with LPS (1 μg/mL, Wako) overnight under 5% CO_2_ at 37 °C, and then subjected to various treatments.

### 2.4. Cytotoxicity Assay

After being treated with EsA, Esg-A, and LPS overnight, DCs were harvested and washed in phosphate-buffered saline (PBS), and suspended in a staining buffer (PBS and 2% FBS, eBioscience, Waltham, MA, USA). Then, DCs were stained with Annexin V-fluorescein isothiocyanate (FITC) and 7-amino actinomycin D (7-AAD) (eBioscience), and measured with a BD Accuri™ Plus Flow Cytometer (BD Biosciences, Franklin Lakes, NJ, USA). Cell viability was analyzed by counting Annexin V- and 7-AAD-negative cells.

### 2.5. Flow Cytometric Cell Surface Staining

After EsA, Esg-A, and LPS treatment, DCs were harvested and resuspended in a staining buffer. DCs were preincubated with anti-mouse CD16/32 (eBioscience). Then, the cell surface staining was performed with the fluorescently labeled antibodies phycoerythrin (PE)-conjugated anti-mouse CD11c, anti-mouse MHC class II-FITC, or anti-mouse CD86-FITC, (all eBioscience) for 30 min at 4 °C. Appropriate isotype controls were used to set the gating on the dot plots. Finally, the double-stained cells were washed and measured using the BD Accuri C6 Plus cytometer.

### 2.6. ELISA Assay for Cytokine Production

DCs were pretreated with EsA or Esg-A for 1 h before stimulation with LPS (1 μg/mL). After 24 h, the culture supernatants were gathered and stored at −80 °C. The secretion of IL-12 and TNF-α was measured using the corresponding mouse ELISA kits (eBioscience).

### 2.7. Endocytosis Assay

After the DCs were preincubated with 1 mg/mL dextran-FITC (42,000 molecular weight; Sigma, St. Louis, MO, USA) at 37 °C for 1 h, the cells were washed, and stained using PE-conjugated anti-mouse CD11c (eBioscience). The endocytosis of DCs were measured using flow cytometry. In addition, parallel experiments were conducted at 4 °C to show that dextran uptake by DCs is suppressed at low temperatures.

### 2.8. Mixed Lymphocyte Reaction (MLR) Assay

CD3+ T lymphocytes from C57BL/6 mouse splenocyte suspensions were isolated by a magnetic antibody cell sorting (MACS) column (Miltenyi Biotec, Bergisch Gladbach, Germany), with greater than 95% purity. After pretreatment with EsA, Esg-A, or LPS overnight, the DCs were pretreated with mitomycin C (80 μg/mL, Sigma) for 1 h and added in graded doses to 5 × 10^5^ CD3+ T lymphocytes in U-bottom 96-well plates. T lymphocyte proliferation during the last 24 h of the 5-day culture was analyzed by MTT assay. The absorbance was read at 570 nm using a microplate reader (Tecan Group Ltd., Maennedorf, Switzerland).

### 2.9. Western Blotting

After pretreatment with EsA, Esg-A, and LPS overnight, DCs were collected and kept at −80 °C. The protein was extracted using RIPA lysis buffer and protease inhibitor, fully lysed by shaking and sonication and centrifuged. The supernatant protein was collected for quantification using a bicinchoninic acid protein assay (Takara Bio, Kusatsu, Japan), and adjusted to the same concentration. Then, the proteins were denatured, and separated by 10% sodium dodecyl sulfate-polyacrylamide gel electrophoresis (Atto, Tokyo, Japan) and semi-dry transferred onto polyvinylidene difluoride membranes (Merck Millipore, Burlington, MA, USA). The membranes were blocked in a blocking reagent (Toyobo, Osaka, Japan), and then incubated overnight in Can Get Signal^®^ (Toyobo, Osaka, Japan) Solution containing the specific primary antibodies anti-TLR4, anti-MyD88, anti-TRAF6, anti-phospho-NFκB (p-NFκB), anti-glyceraldehyde-3-phosphate dehydrogenase (GAPDH), or anti-β-actin (Cell Signaling Technology, Danvers, MA, USA). After washing, the membranes were reacted with horseradish peroxidase-conjugated secondary antibody (Sigma). The blots were developed and detected and then quantified with an iBright System (Thermo Fisher Scientific, Rockford, IL, USA).

### 2.10. Statistical Analysis

The data are shown as the mean ± SEM and analyzed using Prism 8 (GraphPad Software, San Diego, CA, USA). The data were analyzed with Student’s *t*-test. A variance similarity test (f-test) was conducted before the t-test (one-tailed and unpaired), and *p*-values < 0.05 were considered to be significant.

## 3. Results

### 3.1. EsA/Esg-A Cytotoxicity in LPS-Treated Mature DCs

Firstly, various concentrations of EsA and Esg-A were investigated for cytotoxicity to LPS-treated mature DCs using a flow cytometer (Figure 1B). When the alive cell percentage of the control mature DCs (LPS-treated only) was normalized to 1.0, the ratios of alive DCs were 0.97, 0.97, 0.91, and 0.78 in the present EsA at 10, 30, 100, and 300 μM, and 0.98, 1.02, 0.99 and 0.98 in the present Esg-A at 1, 3, 10 and 30 μM, respectively. Further, to analyze EsA/Esg-A-induced apoptosis in mature DCs, the average percentage of apoptotic DCs (Annexin V-FITC positive and 7-AAD negative) was 7.25 for the control, and 8.36, 8.31, 11.24, and 20.6 for treatment with the respective EsA, and 9.52, 5.88, 8.02, and 9.61 with the respective Esg-A. Thus, EsA at 300 μM was found to exhibit cytotoxicity to mature DCs.

### 3.2. EsA/Esg-A Impair Phenotypic Maturation of DCs

We next tested the different concentrations of EsA and Esg-A on DC maturation. In Figure 2A,B, 100 µM of EsA and 10 μM of Esg-A were shown to inhibit CD86 and MHC-II expression in CD11c+ cells. Their inhibitory effects were dose-dependent (Figure 2C), and the 50% inhibition of the increased expression (IC50) of Esg-A is lower than that of EsA. Upon LPS activation, DCs up-regulated co-stimulatory molecule CD86 and large amounts of MHC-peptide complexes within 24 h; treatment with EsA and Esg-A impaired the expression ratios of costimulatory molecule CD86 and MHC class II molecules.

### 3.3. EsA/Esg-A Inhibit LPS-Activated IL-12 and TNF-α Production by DCs

We next tested the DCs’s ability to generate proinflammatory cytokines, since DCs, just like macrophages and monocytes, are reported to be a source of proinflammatory molecules. As shown in Figure 3, EsA and Esg-A inhibited IL-12 and TNF-α production in a concentration-dependent manner. LPS-activated DCs produced higher concentrations of IL-12 and TNF-α than untreated DCs; EsA and Esg-A impaired the secretions of IL-12 and TNF-α in the presence of LPS stimulation, indicating that exposure to EsA and Esg-A impaired the DCs’ capacity to produce proinflammatory cytokines.

### 3.4. EsA/Esg-A Increase Endocytosis of Dextran-FITC in LPS-Activated DCs

Immature DCs have higher endocytic ability, which is lost once DCs become mature. We investigated the ability of EsA- and Esg-A-treated DCs to endocytose dextran. After incubating murine DCs with EsA and Esg-A in the presence of LPS stimuli, dextran-FITC was added to the culture solution. The percentage of double-positive cells (dextran-FITC/CD11c-PE) was not different from that of 100 µM of EsA- or 10 µM of Esg-A-treated DCs to untreated DC. The rate of LPS-activated DCs was lower than that of untreated DCs. However, the EsA- or Esg-A-pretreated DCs exhibited a higher endocytic capacity for dextran-FITC than LPS-stimulated DCs, suggesting that the uptake of dextran was reversed by EsA or Esg-A (Figure 4). The parallel experiment was performed at 4 °C to check the nonspecific uptake of dextran by DCs.

### 3.5. EsA/Esg-A Decrease Allostimulatory Capacity of DCs

Unlike immature DCs, mature DCs promote the proliferation of allogeneic T cells. To test whether EsA and Esg-A impair the maturation of LPS-activated DCs, DCs from day 8 cultures incubated with EsA, Esg-A, or LPS overnight were investigated for their capacity to stimulate allogeneic T cells. In Figure 5, LPS-stimulated DCs exhibited increased proliferative responses relative to untreated DCs, while EsA- or Esg-A-treated DCs exhibited reduced LPS-stimulated proliferative responses. These results indicate that exposure to EsA or Esg-A impaired the allostimulatory capacity of DCs.

### 3.6. EsA/Esg-A Suppress the TLR4-MyD88-NFκB Pathway in LPS-Induced DC Maturation

NFκB activation has an important role in DC maturation. To evaluate the effects on TLR4-MyD88-TRAF6 signaling and NFκB phosphorylation, immature DCs were pretreated by EsA or Esg-A before LPS stimuli. In Figure 6, pretreatment with EsA or Esg-A significantly inhibited the LPS-stimulated up-regulation of TLR4 expression and showed a tendency to attenuate MyD88, TRAF6, and p-NFκB expression.

## 4. Discussion

This study tested the effects of the fresh tomato saponin EsA on innate immunocompetent cells. Using a murine bone marrow-derived DC primary culture and then LPS-stimulated maturation, we showed that EsA and its aglycon Esg-A suppressed DCs’ phenotypic and functional maturation by attenuating TLR4-MyD88-TRAF6 and p-NFκB signaling. Our previous study reported that EsA ameliorated mouse experimental dermatitis [20]. We investigated the underlying mechanism in acquired immunocompetent cells and found that EsA and Esg-A inhibited CD4+ T-cell activation by modulating Th2/Th1/Treg differentiation [23]. Thus, this study shows a new insight into the immunopharmacology by the fresh tomato saponin EsA.

Firstly, after checking the concentration range of EsA/Esg-A without cytotoxicity on mature DCs, we investigated how tomato saponin altered surface-molecule expression by DCs exposed to EsA and Esg-A. We found that EsA and Esg-A inhibited MHC-II and CD86 expressions, and EsA was found to profoundly down-regulate the expression of CD86 costimulatory molecules. In mice, compared with CD80, CD86 deficiency induces a broader impact on antibody class switching, particularly in an adjuvant absence [25]. Our previous study showed in vitro suppression by EsA and Esg-A on the Th2 response, which suggested they prevent Ig class switching to IgE. Further, another study showed that esculeoside B (EsB), a solanocapsine glycoside in juice or canned tomato, ameliorated experimental dermatitis in mice by a decrease in Th2 cytokine production and serum IgE levels [21]. These may support the enhanced CD86 suppression by tomato saponin in LPS-stimulated DCs. Kennedy et al. also reported that CD86 is a more effective CD28 ligand for stimuli-activated T lymphocytes and Treg [26]. Thus, we suggest that EsA and Esg-A could impair the ability of DC to bind to CD28 ligands and consequently the failure of costimulatory signaling to T cells.

Next, we tested the antigen-presenting capacity of DCs with an MLR assay through the T-cell proliferation measurement in the absence or presence of DCs. We found that, after EsA or Esg-A treatment, DCs were poor stimulators of naïve allogeneic T-lymphocyte proliferation, and indeed, it was indicated that DCs’ antigen-presenting function is attenuated by EsA or Esg-A. Moreover, DCs exposed to EsA or Esg-A showed higher endocytic capacity for dextran, again indicating that LPS-induced functional maturation of DCs was suppressed. The important function of DCs is to initiate the immune response by presenting antigens to naïve T cells. Further, DCs are also shown to sustain some chronic inflammatory disorders, for example, allergic diseases, such as atopic dermatitis and rhinitis, hypersensitivity, and collagen diseases. This suggests that antigen-presenting DCs should be an appropriate target for controlling these chronic inflammatory diseases. This finding supports the anti-inflammatory activity of EsA and Esg-A on stimulated DCs, which may contribute to the EsA-mediated alleviation of mouse experimental dermatitis.

Then, we investigated how EsA and Esg-A affect DCs’ ability to produce proinflammatory cytokines in response to LPS stimuli. IL-12 is a more specific signal in functionally activated DCs because IL-12 will lead to the effector T-cell expansion, induce Th1 response, and activate proinflammatory cytokine production. Additionally, TNF-α is mainly induced by DCs and regulates their function and development [27]. In this study, DCs produced large amounts of IL-12 and TNF-α after LPS stimulation, while EsA- or Esg-A-treated DCs produced markedly less IL-12 and TNF-α. This finding may suggest the anti-inflammatory mechanism of EsA and Esg-A because proinflammatory cytokines may support the immune reaction at inflammatory lesions. Additionally, lower IL-12 production in DCs subsequently could result in the suppression of Th1 differentiation, which is supported by our previous study on CD4 T cells [21].

To investigate the intracellular mechanisms responsible for suppressed phenotypic and functional maturation of DCs by EsA or Esg-A, we examined LPS-stimulated TLR4-MyD88-TRAF6 signaling and p-NFκB. NFκB activation regulates the expression of proinflammatory cytokine genes and has an important role in DC maturation [28,29]. However, in this study, we provided evidence that the inhibitory effect of EsA and Esg-A on DC maturation involves TLR4 inhibition, and is associated with the attenuation of MyD88, TRAF6, and p-NFκB signaling.

Moreover, in comparing the inhibitory potential of EsA and Esg-A in LPS-induced DC activation, the inhibition by Esg-A was found to be greater than that by EsA, suggesting Esg-A as an active moiety on such murine DC activation. Thus, it is necessary to investigate the pharmacokinetics of fresh tomato saponin.

## 5. Conclusions

These data show that saponin EsA and its aglycon Esg-A from fresh tomatoes are capable of attenuating in vitro LPS-stimulated phenotypic and functional maturation of murine DCs via down-regulating the TLR4-MyD88-TRAF6 and p-NFκB signaling pathways. Thus, the underlying mechanism of anti-allergy and anti-inflammatory activity by EsA includes immunoregulatory functions on both adaptive and native immunity. The next steps include an investigation of the effect of fresh tomato saponin on regulatory DC differentiation, and the effects of IL-10 and TGF-β on regulatory DCs and Treg cells.

## Figures and Tables

**Figure 1 nutrients-16-01699-f001:**
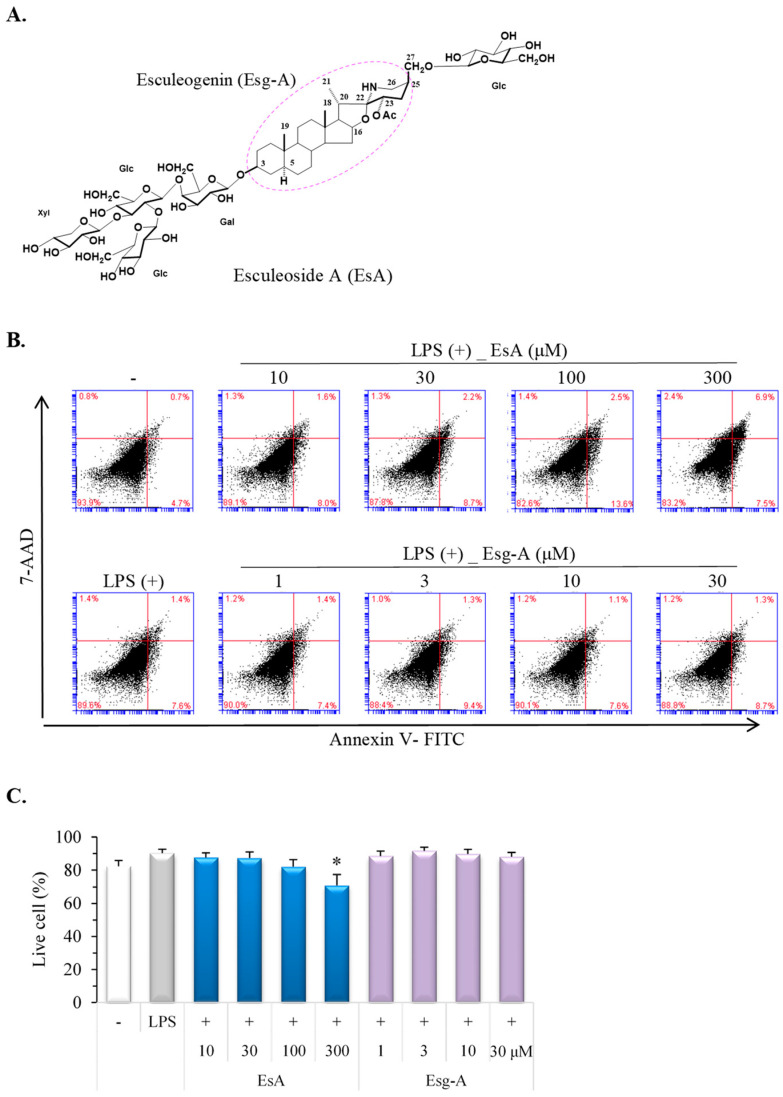
EsA/Esg-A cytotoxicity in murine bone marrow-derived DCs. (**A**) EsA and Esg-A chemical structures. EsA: esculeoside A; Esg-A: esculeogenin A; Gal: galactose; Glc: glucose; Xyl: xylose. Molecular weight: 1270.38 (EsA); 447.66 (Esg-A). (**B**) Immature DCs were produced from mouse bone marrow cells using a treatment of 2 ng/mL of GM-CSF for 8 days. Immature DCs were pretreated with 0.1% DMSO, or the indicated EsA or Esg-A concentrations for 1 h, and then LPS (1 μg/mL) stimuli for 24 h. Cell viability was investigated using Annexin V-FITC and 7-AAD by a flow cytometer. (**C**) The results represent 5 independent experiments. *: *p* < 0.05, significantly different from the LPS alone (control).

**Figure 2 nutrients-16-01699-f002:**
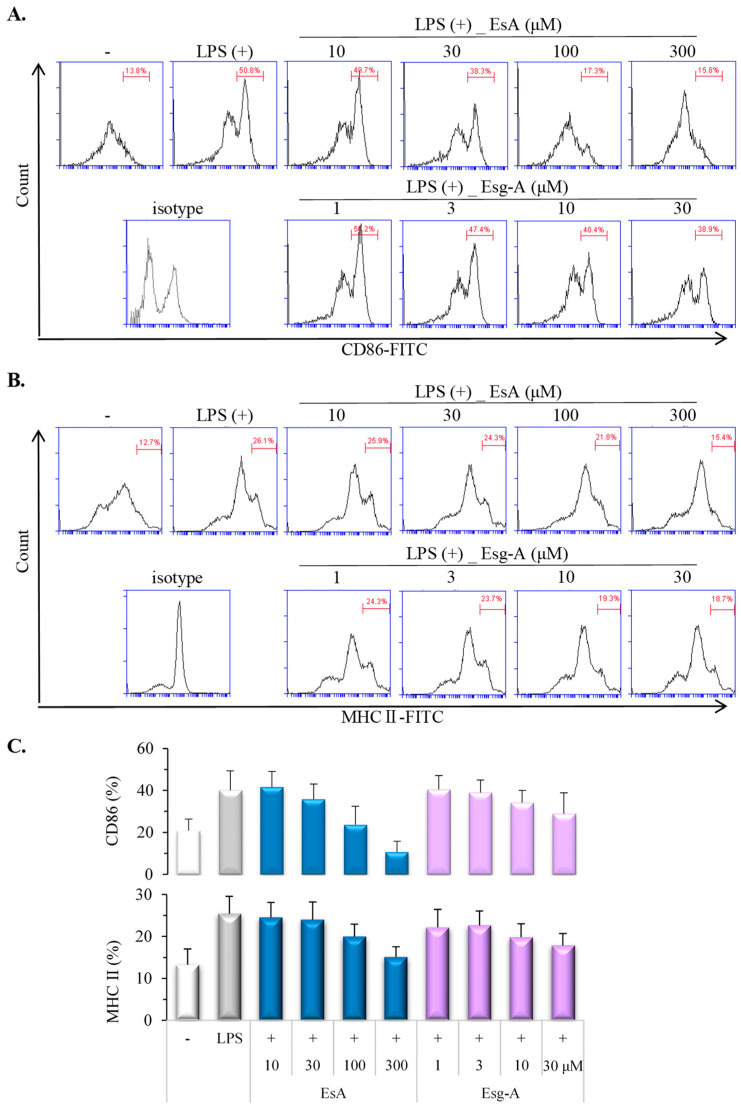
EsA/Esg-A inhibited phenotypic activation of DCs. Immature DCs were produced from mouse bone marrow cells by GM-CSF treatment. After harvesting, immature DCs were pretreated by 0.1% DMSO, or EsA or Esg-A for 1 h, and then LPS stimuli overnight. DCs were stained with PE-conjugated anti-CD11c antibody, and the phenotypic expression level was measured using CD86-FITC or MHC-II-FITC and then investigated by double-stained flow cytometry. (**A**) Representative histogram plots show the expression of costimulatory molecule CD86 during DC maturation without and with EsA/Esg-A, while the isotype control shows untreated DCs. (**B**) Representative histogram plots show the expression of MHC-II during DC maturation without and with EsA/Esg-A, while the isotype control shows untreated DCs. (**C**) Expression ratios of CD86 and MHC-II during DC maturation without and with EsA or Esg-A. Data show five independent experiments.

**Figure 3 nutrients-16-01699-f003:**
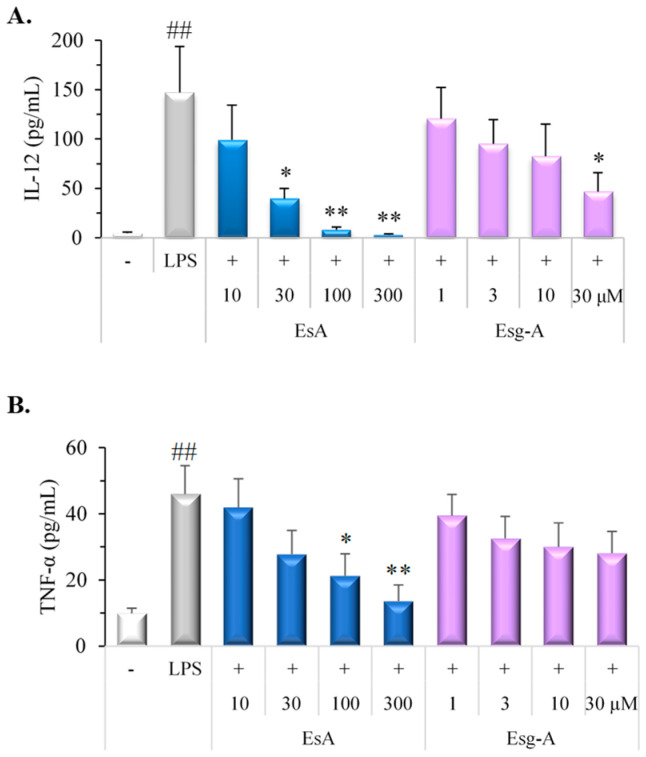
EsA/Esg-A inhibited cytokine production by LPS-activated DCs. Immature DCs were generated from mouse bone marrow cells by GM-CSF treatment. After harvesting, immature DCs were pretreated with DMSO, EsA, or Esg-A and then stimulated with LPS overnight. The culture supernatants were gathered and assayed for cytokine secretion by ELISA. (**A**) IL-12 production levels during DC maturation without and with EsA/Esg-A. (**B**) Production levels of TNF-α during DC maturation without and with EsA or Esg-A. Data show five independent experiments. *: *p* < 0.05, **: *p* < 0.01, significantly different from the control (LPS alone). ##: *p* < 0.01, significantly different from the no LPS stimuli.

**Figure 4 nutrients-16-01699-f004:**
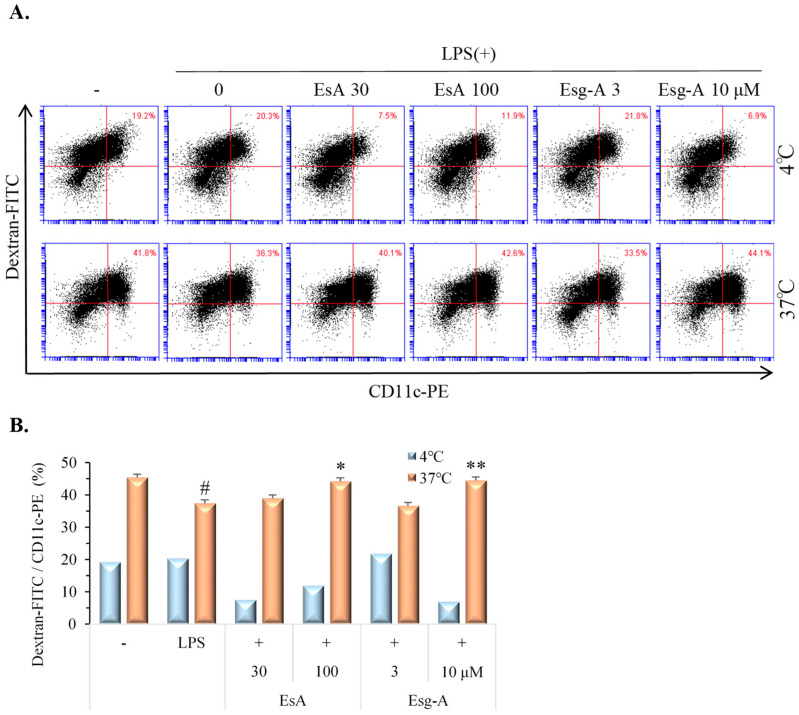
EsA/Esg-A increased antigen uptake in LPS-stimulated DCs. Immature DCs were produced from mouse bone marrow cells by GM-CSF treatment. After harvesting, immature DCs were pretreated with DMSO, EsA (30, 100 μM), or Esg-A (3, 10 μM) for 1 h, and then stimulated with LPS (1 μg/mL) overnight. DCs were incubated with dextran-FITC (1 mg/mL) for 1 h at 4 °C or 37 °C. After washing, DCs were stained using PE-conjugated anti-CD11c, and double-stained DCs were measured by flow cytometry. (**A**) Representative dot plots show the ratio of dextran-FITC+/CD11c-PE+ DCs during maturation without and with EsA/Esg-A. The control endocytic activity was measured after treatment with dextran-FITC at 4 °C. (**B**) Endocytic activity of DCs during maturation without and with EsA or Esg-A. Data show three independent experiments. *: *p* < 0.05, **: *p* < 0.01, significantly different from the control (LPS alone) at 37 °C. #: *p* < 0.05, significantly different from the no LPS stimuli.

**Figure 5 nutrients-16-01699-f005:**
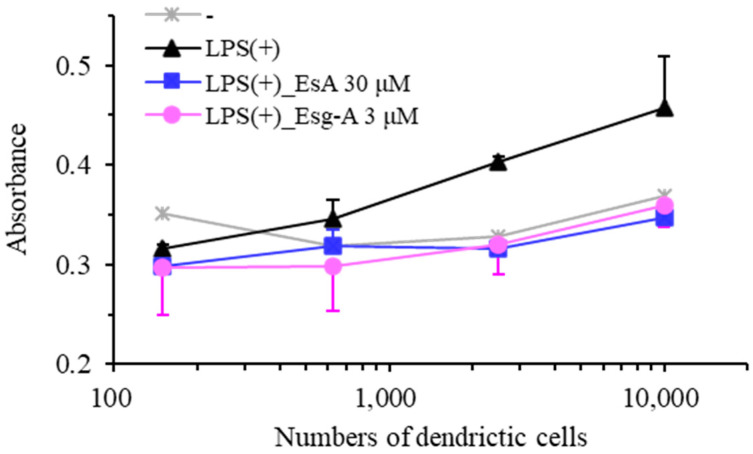
EsA/Esg-A decreased proliferation of allogeneic T cells through the maturation of DCs. Immature DCs were produced from mouse bone marrow cells using GM-CSF treatment. After harvesting, immature DCs were treated with 0.1% DMSO, or 30 μM of EsA or 3 μM of Esg-A for 1 h, and then stimulated with LPS (1 μg/mL) overnight. The treated DCs were harvested and washed extensively to remove EsA/Esg-A, then incubated with 80 μg/mL of mitomycin C for 1 h. Mitomycin-treated DCs were added in graded densities to 5 × 10^5^ T cells purified from C57BL/6 mice. After culture for 4 days, the T-cell proliferation was analyzed with an MTT assay. Data represent three independent experiments.

**Figure 6 nutrients-16-01699-f006:**
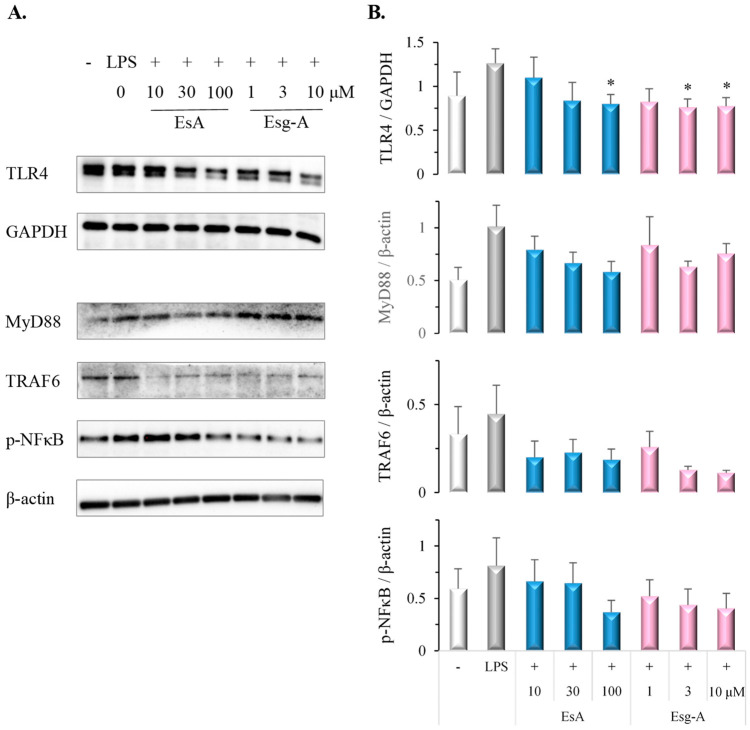
EsA/Esg-A inhibited TLR4 signaling in LPS-stimulated DCs. Immature DCs were produced from mouse bone marrow cells by GM-CSF treatment. After harvesting, immature DCs were pretreated with DMSO, EsA, or Esg-A for 1 h, and then stimulated with LPS overnight. Total cell extracts were prepared and blotted with anti-TLR4, anti-TRAF6, anti-MyD88, and anti-p-NFκB antibodies. (**A**) Representative bands show TLR4, TRAF6, MyD88, and p-NFκB protein expressions during DC activation without and with EsA/Esg-A. (**B**) TLR4, TRAF6, MyD88, and p-NFκB band intensities were normalized to GAPDH or β-actin protein expression, respectively. Data represent four independent experiments. *: *p* < 0.05, significantly different from LPS alone (control).

## Data Availability

Data are contained within the article.

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
