# Peer review of "Saponin Esculeoside A and Aglycon Esculeogenin A from Ripe Tomatoes Inhibit Dendritic Cell Function by Attenuation of Toll-like Receptor 4 Signaling"

_nutrients, 2024, doi:10.3390/nu16111699_

Round 1

Reviewer 1 Report

Comments and Suggestions for Authors

The study by Zhou et al. describes effects of ripe tomato saponin on LPS-induced murine BM-derived dendritic cell maturation. Of note, these effects are quite remarkable, since the investigated compounds EsA and Esg-A markedly inhibit all analyzed markers and functions of activation/maturation induced by LPS. However, the basis or mechanisms of these effects still remain enigmatic.

Major comments

Generally, I am missing the effects of EsA and Esg-A on unstimulated (no LPS) DCs. Particularly for the endocytosis assay as well as the analysis of signaling this would be crucial to obtain a view on the mechanism of action.

Is the downregulation of endocytic receptors blocked by the tomato saponins? This would be an essential experiment. How can the large differences at 4° be explained?

Fig.6. Classically I would expect earlier signaling events to be analysed for proving a direct intervention with the TLR4 pathway such as MAPK phosphorylations, IkappaB degradation, NFkappaB translocation to the nucleus. Whole cell expression of signaling molecules after 24 hours may not be the best choice. In case I am wrong here, please give respective references! Anyway, as mentioned before, samples without LPS would be crucial here to pinpoint the level at which the saponins interfere with DC maturation.

Minor

The cytotoxic and apopototic action of EsA, though visible only at higher concentrations, should not be left uncommented in the discussion.

Please comment on the differences between EsA and Esg-A action and what´s the rationale for the comparison of these two compounds.

Author Response

Dear Reviewer,

Thank you for your review.

Major comments

  1. Generally, I am missing the effects of EsA and Esg-A on unstimulated (no LPS) DCs. Particularly for the endocytosis assay as well as the analysis of signaling this would be crucial to obtain a view on the mechanism of action.

Atopic dermatitis lesions were characterized by expanded not only type 2 T cells but also inflammatory DCs (Single-cell transcriptome analysis of human skin identifies novel fibroblast subpopulation and enrichment of immune subsets in atopic dermatitis. He H, Suryawanshi H, Morozov P, Gay-Mimbrera J, Del Duca E, Kim HJ, Kameyama N, Estrada Y, Der E, Krueger JG, Ruano J, Tuschl T, Guttman-Yassky E. J Allergy Clin Immunol. 2020 Jun;145(6):1615-1628. doi: 10.1016/j.jaci.2020.01.042.), therefore, to mimic the response of inflammatory DCs, LPS-stimulated DCs were used in the present study.

A reference is added on page 2, line 53, and page 13, line 396 as [22].

Of course, it is important to investigate the effects of EsA and Esg-A on immature DCs in the next step.

  1. Is the downregulation of endocytic receptors blocked by the tomato saponins? This would be an essential experiment. How can the large differences at 4° be explained?

The present data shows that tomato saponin maintained DC endocytic activity. Although endocytosis is receptor-mediated, it is hard to say if the downregulation of the endocytic receptor is directly blocked by tomato saponin.

To check the nonspecific uptake of dextran by DCs, the parallel experiment was done at 4 °C. Perhaps our temperature control is not so good, thus may induce a little large difference at 4 °C.

  1. Figure 6. Classically I would expect earlier signaling events to be analysed for proving a direct intervention with the TLR4 pathway such as MAPK phosphorylations, IkappaB degradation, NFkappaB translocation to the nucleus. Whole cell expression of signaling molecules after 24 hours may not be the best choice. In case I am wrong here, please give respective references! Anyway, as mentioned before, samples without LPS would be crucial here to pinpoint the level at which the saponins interfere with DC maturation.

Thank you for your advice. Because there are fewer mice primary DCs, we haven’t succeeded in cytosol and/or nuclear expression of signaling molecules.

TLR4 is activated by LPS, and sequentially triggers two signaling cascades: the first involving adaptor protein MyD88 (myeloid differentiation primary response gene) is induced in the plasma membrane, whereas the second engaging adaptor protein TRIF begins in early endosomes ([8] Kuzmich et al. TLR4 Signaling Pathway Modulators as Potential Therapeutics in Inflammation and Sepsis. Vaccines (Basel). 2017, 5, 34. doi: 10.3390/vaccines5040034), then MyD88-TRAF6 (TNF receptor-associated factor 6), parts of a submembranous signaling complex, induce the activation MAPK and IKK (page 1, line 37). Thus, TLR4-MyD88-TRAF6 is considered the earlier signaling event.

As you mentioned, it is interesting to investigate the effect of tomato saponin on the distinct stages of DCs (pre-DC, immature, and mature DCs) in the next step.

Minor

  1. The cytotoxic and apopototic action of EsA, though visible only at higher concentrations, should not be left uncommented in the discussion.

We checked the concentration range of EsA/Esg-A without cytotoxicity on DCs. This sentence is added on page 10, line 276 in the discussion.

  1. Please comment on the differences between EsA and Esg-A action and what´s the rationale for the comparison of these two compounds.

In comparing EsA- and Esg-A-inhibitory potential in LPS-induced DC activation, the inhibition by Esg-A is greater than that by EsA with glycosteroid structure, suggesting that the Esg-A moiety may be mainly responsible for the inhibitory effect on such murine DC activation, thus Esg-A with steroid structure should be an active moiety. Such main effect of Esg-A is consistent with our previous report because, in the urine of men orally administered ripe tomato fruit, the final metabolites were eliminated as androsterone analogues [21]. (Page 11, line 322)

Best Regards,

Jian-Rong Zhou

Reviewer 2 Report

Comments and Suggestions for Authors

It is an interesting paper on comprehensive studies on the biologial effects of cherry tomato fruitc natural components - esculeoside A and its aglycon - esculeogenin A on mice dendritic cells. The studies had shown that beneficial effect of both compounds and their possible use in immunopharmacology. Paper is well composed and written. The latin name of tomato shopuld be written in italics, but this error could be corrected at the stage of proofreading.

Author Response

Dear Reviewer,

Thank you for your review.

  1. It is an interesting paper on comprehensive studies on the biologial effects of cherry tomato fruitc natural components - esculeoside A and its aglycon - esculeogenin A on mice dendritic cells. The studies had shown that beneficial effect of both compounds and their possible use in immunopharmacology. Paper is well composed and written. The latin name of tomato shopuld be written in italics, but this error could be corrected at the stage of proofreading.

The Latin name of tomato has been written in italics on page 2, line 63.

Best Regards

Jian-Rong Zhou

Reviewer 3 Report

Comments and Suggestions for Authors

The present manuscript reports the immunoregulatory activity on DCs of the tomato saponin Esculeoside A (EsA) and its aglycone Esg-A. The study aims to address the mechanism of action of these molecules after previous experiments showing that the glycosteroid can improve atopic dermatitis in mice. The manuscript is interesting and the data appear to be consistent with in vivo observations. However, the texts is poorly written with many inconsistencies between the data and their discussion. Furthermore, there are some points that are not clear at all.

Figure 1 – It is not clear whether the data on apoptotic activity at 300 ug/ml are significant. A statistical analysis is mandatory.

Figure 2 – there are more than one inconsistency between the facs plots and the bar chart below. In particular, the numbers reported in the plots are very different from those reported in the graphs. 

Figure 2 – According to the plots, the effect of Esg-A on MHCII is non-linear and there is an increase in the expression of the surface marker at 30 ug/ml. 

Other points that should be considered:

The addition of EsA or Esg-A could influence the release of IL-10. Can the author measure this cytokine in the media of their experiments?

In relation to the reduction of CD3 proliferation by EsA or Esg-A, can the authors define whether the effect is due to the decrease in CD4 or CD8 T cells? This information would be very important to define the mechanism of cross-presentation. In the discussion, the authors report CD4+ inhibition from a previous article (reference 22) but the cited article does not show this effect.

In this version the manuscript is not suitable for publication. I suggest reanalyzing the experiments and rewriting the manuscript paying more attention to a rigorous discussion of the data.

Comments on the Quality of English Language

None

Author Response

Dear Reviewer,

Thank you for your review.

  1. Figure 1 – It is not clear whether the data on apoptotic activity at 300 ug/ml are significant. A statistical analysis is mandatory.

The cytotoxicity at 300 ug/ml is significant (p<0.05). The * symbol is added. Apoptotic activity is changed as cytotoxicity (page 4, line 158).

  1. Figure 2 – there are more than one inconsistency between the facs plots and the bar chart below. In particular, the numbers reported in the plots are very different from those reported in the graphs.

We made a mistake, and the bar graph of CD86 is corrected in Figure 2C.

  1. Figure 2 – According to the plots, the effect of Esg-A on MHCII is non-linear and there is an increase in the expression of the surface marker at 30 ug/ml.

We made a mistake. It is corrected.

Other points that should be considered:

  1. The addition of EsA or Esg-A could influence the release of IL-10. Can the author measure this cytokine in the media of their experiments?

We have not measured the release of IL-10. It is interesting to know such an effect by EsA or Esg-A in DCs.

  1. In relation to the reduction of CD3 proliferation by EsA or Esg-A, can the authors define whether the effect is due to the decrease in CD4 or CD8 T cells? This information would be very important to define the mechanism of cross-presentation. In the discussion, the authors report CD4+ inhibition from a previous article (reference 22) but the cited article does not show this effect.

In the discussion, we have written that EsA and Esg-A could inhibit CD4+ T cell activation by modulation of Th2/Th1/Treg differentiation (page 10, line 273), and could result in the suppression of the Th1 differentiation (page 11, line 313), these did not mean a direct decrease in the proportions of CD4+ T cells, which is from our previous article.

Our previous report found the reduction of ConA-blast CD3 proliferation by EsA or Esg-A, the present study shows EsA or Esg-A impaired the allostimulatory capacity of DCs, where CD3+ T cells were used. Thus, it is still hard to define whether the effect is due to CD4 and/or CD8 T cells.

  1. In this version the manuscript is not suitable for publication. I suggest reanalyzing the experiments and rewriting the manuscript paying more attention to a rigorous discussion of the data.

We reanalyzed some data and corrected the parts of our manuscript.

Best Regards,

Jian-Rong Zhou

Round 2

Reviewer 1 Report

Comments and Suggestions for Authors

The replies to comments 1-3 are not satisfactory. I can understand that the number of animals and cells from these animals are limited and not every experiment can be repeated with the demanded controls. However, since all experiments assess effects of an overnight to 24h stimulation with LPS in presence of the investigated components, it is essential to show the effects of these components alone, without stimulation.

Author Response

Dear Reviewer,

Thank you for pointing it out.

The replies to comments 1-3 are not satisfactory. I can understand that the number of animals and cells from these animals are limited and not every experiment can be repeated with the demanded controls. However, since all experiments assess effects of an overnight to 24h stimulation with LPS in presence of the investigated components, it is essential to show the effects of these components alone, without stimulation

It may be a similar cytotoxicity by EsA / Esg-A in murine-derived primary immune cells, thus we show the effects of overnight to 24 h without stimulation in the presence of EsA or Esg-A alone on mice primary T cells in Supplement Figure 1.

Of course, it is important to investigate the effects of EsA and Esg-A on immature DCs in the next step.

Best Regards,

Jian-Rong Zhou

Reviewer 3 Report

Comments and Suggestions for Authors

I am unable to detect the changes made by the authors.

My request of additional data on cytokine production (i.e., IL-10) has not been addressed while I believe it would be very important to support the discussion and conclusions.

Comments on the Quality of English Language

No comments

Author Response

Dear Reviewer,

Thank you for pointing it out.

My request of additional data on cytokine production (i.e., IL-10) has not been addressed while I believe it would be very important to support the discussion and conclusions.

Our previous study demonstrated that EsA/Esg-A suppressed IL-10 secretion and Foxp3 gene expression in concanavalin A-blast T cell proliferation. Recently, we found EsA/Esg-A increased IL-10 production in weak stimuli to T cells (Unpublished data). Thus, we hope it will be carefully discussed on the production of Treg cytokine IL-10 because we haven’t enough data at the present DCs. Of course, it is important to investigate the effects of EsA and Esg-A on cytokine IL-10 production of DCs in the next step.

Best Regards,

Jian-Rong Zhou